# Reovirus Type 3 Dearing Variants Do Not Induce Necroptosis in RIPK3-Expressing Human Tumor Cell Lines

**DOI:** 10.3390/ijms24032320

**Published:** 2023-01-24

**Authors:** Diana J. M. van den Wollenberg, Vera Kemp, Martijn J. W. E. Rabelink, Rob C. Hoeben

**Affiliations:** Department of Cell and Chemical Biology, Leiden University Medical Center, 2300 RC Leiden, The Netherlands

**Keywords:** reovirus, necroptosis, mixed-lineage kinase domain-like protein, receptor-interacting protein kinase 3

## Abstract

Reoviruses are used as oncolytic viruses to destroy tumor cells. The concomitant induction of anti-tumor immune responses enhances the efficacy of therapy in tumors with low amounts of immune infiltrates before treatment. The reoviruses should provoke immunogenic cell death (ICD) to stimulate a tumor cell-directed immune response. Necroptosis is considered a major form of ICD, and involves receptor-interacting protein kinase 1 (RIPK1), RIPK3 and phosphorylation of mixed-lineage kinase domain-like protein (MLKL). This leads to cell membrane disintegration and the release of damage-associated molecular patterns that can activate immune responses. Reovirus Type 3 Dearing (T3D) can induce necroptosis in mouse L929 fibroblast cells and mouse embryonic fibroblasts. Most human tumor cell lines have a defect in RIPK3 expression and consequently fail to induce necroptosis as measured by MLKL phosphorylation. We used the human colorectal adenocarcinoma HT29 cell line as a model to study necroptosis in human cells since this cell line has frequently been described in necroptosis-related studies. To stimulate MLKL phosphorylation and induce necroptosis, HT29 cells were treated with a cocktail consisting of TNFα, the SMAC mimetic BV6, and the caspase inhibitor Z-VAD-FMK. While this treatment induced necroptosis, three different reovirus T3D variants, i.e., the plasmid-based reverse genetics generated virus (T3D^K^), the wild-type reovirus T3D isolate R124, and the junction adhesion molecule-A-independent reovirus mutant (jin-1) failed to induce necroptosis in HT29 cells. In contrast, these viruses induced MLKL phosphorylation in murine L929 cells, albeit with varying efficiencies. Our study shows that while reoviruses efficiently induce necroptosis in L929 cells, this is not a common phenotype in human cell lines. This study emphasizes the difficulties of translating the results of ICD studies from murine cells to human cells.

## 1. Introduction

In a tumor, cancer cells are surrounded by a heterogeneous population of other cell types, together composing the tumor microenvironment (TME). The composition of this TME is complex and differs by tumor type. The communication within the network of cells in the TME is very dynamic and consists of many factors, like chemokines, cytokines, growth factors, proteases and other enzymes present in the tissue surrounding the tumor [1,2]. The TME is an important factor that affects the efficacy of oncolytic virus treatment and other cancer therapies, including immunotherapy [3,4]. Oncolytic virotherapy not only destroys tumor cells as a result of virus replication, but also induces an immune response against cancer cells. Even though this immune response may be directed initially towards the virus-infected cells it can make a tumor more sensitive to additional immunotherapies [5,6,7,8]. For successful treatment, it is important to understand the responses of the tumor and cell types to oncolytic viruses. While there are several pathways leading to virus-induced cell death, not all types lead to the effective induction of an immune response against the cancer cells. Furthermore, many tumors are suppressing detection by or activation of the immune system [9,10].

One frequently used oncolytic virus is the Mammalian Orthoreovirus (reovirus). The Type 3 Dearing (T3D) isolate of this species is currently tested in various clinical trials, both as monotherapy and in combination with other products [11,12]. Reoviruses are non-enveloped viruses with 10 dsRNA genome segments that have a preference for lytic infection of tumor cells and not in diploid normal cells. How the reoviruses cause cell death in a tumor cell depends on the tumor cell type and the pathways that are operational in these cells [13]. Many transformed cells have aberrant signaling pathways to avoid cell death and prevent detection by the immune system. Reoviruses and other oncolytic viruses use these aberrations for their benefit to replicate and produce new progeny [14]. In normal cells, reovirus replication is inhibited by the activation of dsRNA-activated protein kinase R (PKR). In cancer cells with an activated RAS pathway, phosphorylation of PKR is inhibited, creating an opportunity for reoviruses to replicate and induce apoptosis [15]. The contribution of RAS is much studied in the oncolysis of reovirus, but cancer cells with other mutations can also be susceptible to reovirus-induced cell death [16,17].

The most studied cell death pathway related to reovirus oncolysis is the apoptotic pathway [18]. Recent data on oncolytic cell death indicate that reoviruses can also induce a more immunogenic form of cell death called necroptosis [14,19,20]. In contrast to classical extrinsic apoptosis, necroptosis causes the cell membranes to disintegrate and release damage-associated molecular patterns (DAMPs) in the surrounding microenvironment. These DAMPs trigger the innate immune response which may be helpful for immunotherapies. In both apoptosis and necroptosis pathways, caspase 8 plays an important role. Caspase 8 in its active form stimulates executioner caspases 3, 6 and 7 leading to apoptosis. In its inactive form, caspase 8 no longer inhibits phosphorylation of receptor-interacting protein kinase 1 (RIPK1), leading to the attraction of RIPK3, which auto-phosphorylates and in turn attracts and phosphorylates mixed-lineage kinase domain-like protein (MLKL). A complex consisting of oligomers of phosphorylated MLKL (p-MLKL) subsequently translocates to the cell membrane and causes cell disruption [21,22]. A simplified cartoon is shown in Figure 1, in which only the key players in our study are drawn. MLKL and its’ functions in cancer are reviewed in detail by Martens et al. [23]. 

Murine L929 cells are frequently used as a model system to study necroptosis induction by reoviruses. For knockdown or knockout studies related to key players in necroptosis studies, mouse embryonic fibroblast (MEF) cell lines are often used [24,25]. In L929 cells, all key components for necroptosis are expressed and cells are sensitive to this form of cell death. Furthermore, the ability to induce necroptosis is reovirus strain-specific [20,26]. Reovirus T3D can induce cell death in the presence of caspase 8 inhibitor Z-IETD-FMK and the pan-caspase inhibitor Z-VAD(OMe)-FMK (Z-VAD), suggesting a caspase-independent cell death program. Inhibition of RIPK1 by necrostatin-1 results in decreased cell death and delayed loss of intracellular ATP levels, compared to Z-IETD-FMK or non-treated L929 cells. Reovirus type 1 Lang (T1L) is also capable of induction of necroptosis, but the process is slower than upon infection with T3D [26]. 

In the current study, we tested the ability of three reovirus strains (viz. our wild-type T3D isolate R124 (wt-R124), the plasmid-derived reovirus T3D^K^, and our mutant reovirus jin-1) to induce necroptosis in human tumor cell lines. We choose these three viruses because we observed differences in the induction of p-MLKL levels between the viruses in L929 cells. We hypothesized that at least one of the viruses was capable of inducing necroptosis in human cell lines. In human tumor cells, the necroptosis pathway is often downregulated due to the absence of RIPK3 expression [27]. Therefore, we first tested, based on Koo et al., whether our tumor cell lines were capable of inducing p-MLKL levels after treatment with a cocktail of TNFα, SMAC mimetic BV6 and Z-VAD (TBZ). We included the TBZ-sensitive human HT29 cells as a control since this cell line is mostly used as a human model for necroptosis studies. In our panel, we found one additional cell line, STA-et2.1, that could induce p-MLKL expression after treatment with the TBZ cocktail. Surprisingly, while some of the reoviruses (T3D^K^ and jin-1) can induce a strong p-MLKL expression in murine L929 cells, neither was able to induce p-MLKL in the human HT29 and STA-et2.1 cells. The introduction of human RIPK3 in RIPK3-negative human HER911 cells triggers an over-expression of p-MLKL but does not cause cell death in these cells. This shows that human cells can behave differently than murine cells concerning the regulation of necroptosis. This should be considered in necroptosis studies, especially in the context of oncolytic reovirus infection.

## 2. Results

### 2.1. Detection of p-MLKL Induction in Reovirus-Infected Mouse Cells

To establish that our set of viruses could induce necroptosis in mouse L929 cells, we used a Western blot assay. L929 cells were exposed to reoviruses (R124, T3D^K^ and jin-1) or, as a positive control, treated with the chemical cocktail consisting of TNFα, SMAC mimetic BV-6 and Z-VAD (TBZ). Not all viruses showed similar induction of p-MLKL expression in L929 cells, with wt-R124 being the weakest inducer of p-MLKL with only 1.28-fold induction of p-MLKL over mock-treated cells. T3D^K^ was the strongest inducer with 4.6-fold induction (Figure 2A). In the combined analysis of biological and technical replicates, wt-R124 did not significantly increase the MLKL phosphorylation compared to the mock at the chosen time point (30 h), while both T3D^K^ and jin-1 increased the induction of p-MLKL levels significantly. The L929 cells displayed a high percentage of σ3-positive cells upon exposure to any of the three viruses (Appendix A). In the murine cancer-associated fibroblast cells, mCAF-3, all three reoviruses increased p-MLKL, although to lower levels than seen in the L929 cells (Figure 2B). What is noticeable in this cell line is that even though there is less reovirus protein σ3 detected in the T3D^K^-infected cells than in both wt-R124 and jin-1 infected cells, there is a 2.66-fold induction of p-MLKL (Figure 2B). This suggests that reovirus T3D^K^ is a potent inducer of p-MLKL in the mCAF-3 cells. This corresponds with the results of the viability assay. In L929 cells all three viruses cause cell death to the same extent, three days post-infection. Although the three viruses cause cell death in mCAF-3 cells, the T3D^K^ virus is not as effective as the other two (Figure 2C). The amount of σ3-positive cells in T3D^K^-infected mCAF-3 cells is in line with this, showing that T3D^K^ infection is less efficient in these cells compared to jin-1 infection (Appendix A). 

In both murine cell lines, T3D^K^ and to a lesser extent jin-1 can induce p-MLKL. The wt-R124 is a bit less potent in the upregulation of p-MLKL in the L929 cells. These results suggest that the reovirus strain as well as the cell type affect the induction of MLKL phosphorylation.

### 2.2. Induction of p-MLKL in Human Cell Lines

In human tumor cell lines, the necroptosis pathway is often inhibited due to the loss of RIPK3 expression [27]. Therefore, we tested in a small panel of cell lines whether p-MLKL is induced upon stimulation with TBZ. Only one cell line, the osteosarcoma cell line STA-et2.1, showed ~10-fold induction of p-MLKL over mock-treated cells, and in three other cell lines, no induction was observed (Figure 3A). 

Next, we tested the three reoviruses in the STA-et2.1 and HT29 cell lines. We included HT29 cells because they are known to induce p-MLKL upon TBZ treatment. In both cell lines, the TBZ cocktail induced p-MLKL expression. In contrast, none of the reoviruses was able to increase p-MLKL levels (Figure 3B). In both cell lines, there is expression of RIPK3, and this expression is not decreased upon infection by the viruses, suggesting that RIPK3 is still present in infected cells and blockade of p-MLKL expression was not caused by downregulation of RIPK3 expression. Both cell lines are sensitive to reovirus-induced cell death (Figure 3C) although less than the L929 cells. We did check in HT29 cells if caspase 3/7 activity is increased after exposure to the reoviruses. HER911 cells were taken along as a control since they are very sensitive for induction of caspase 3/7 by the three reoviruses. Indeed, both T3D^K^ and jin-1 are potent inducers of caspase 3/7 activity in the HT29 cells in contrast to wt-R124 (not even with an MOI 10), while in HER911 cells all three reoviruses induce similar levels of caspase 3/7 induction (Appendix A). HT29 and STA-et2.1 cells were efficiently infected by all three viruses as demonstrated by the percentage of σ3-positive cells by flow cytometry (Appendix A).

To summarize the results, STA-et2.1 and HT29 cells are sensitive to reovirus-induced cell death by each of the three reoviruses, but the cell death is not caused by an increase in p-MLKL expression in the cell lines. This implies that reoviruses explore other means to induce cell death in these cell lines.

### 2.3. Overexpression of Human RIPK3 in HER911 Cells Does Not Lead to the Induction of p-MLKL by Reoviruses

Since the HER911 cells do not express detectable RIPK3 expression we wondered if the forced expression of *human RIPK3* in HER911 cells would suffice for the induction of necroptosis. To study this, we generated a HER911hRIPK3 polyclonal cell line (HER911hRIPK3 PC100) through lentiviral transduction. We confirmed that the HER911hRIPK3 cells expressed high amounts of RIPK3 mRNA, even more than HT29 and STA-et2.1 cells (Figure 4A). In most of the other tumor cell lines (viz. U118, H1299, A549, HT1080 and U2OS) no RIPK3 gene expression was detected. Only in a normal fibroblast cell line (VH10), RIPK3 mRNA was detected, but these cells resist reovirus infection [28].

We noticed that forced expression of RIPK3 cDNA led to constitutive p-MLKL expression in untreated cells already, whereas p-MLKL is nearly absent in wild-type (WT) HER911 cells (Figure 4B). Importantly, infection with any of the three reoviruses did not markedly increase this basal p-MLKL level. Additionally, treatment with TBZ did not lead to cell death in both HER911 WT and HER911hRIPK3 cells, as is shown by the low fraction of Propidium Iodide (PI)-positive cells after treatment for 24 h (Figure 4C and Appendix A). However, in cell lines HT29, STA-et2.1 and L929, there is an increase in PI uptake after treatment with TBZ suggesting that the cells lose their membrane integrity and PI can enter the cells. Therefore, we believe that in HER911 cells induction of necroptosis is inhibited by other mechanisms than solely the absence of RIPK3 and the consequential absence of p-MLKL.

## 3. Discussion

Necroptosis is considered a form of ICD and is mostly studied in the context of mouse models (both in vivo and in vitro). To explore the possibility of combining oncolytic virotherapy with immunotherapies, it would be useful to know whether human tumor cells harbor a functional necroptotic pathway to facilitate ICD and boost immune responses against the tumor. Here, we show that even though the human cell lines HT29 and STA-et2.1 can upregulate p-MLKL expression upon TBZ treatment, p-MLKL induction is not observed in these cells upon infection with the three reoviruses (wt-R124, T3D^K^ and jin-1) tested (Figure 3 and Figure 4). In the mouse L929 cell line, mainly T3D^K^ and jin-1 induce p-MLKL expression at 30 h post infection, while for wt-R124 p-MLKL expression is increased slightly above the background level (Figure 2). In the murine cancer-associated fibroblasts mCAF-3, all three viruses show a similar induction of p-MLKL expression (between 2.5 and 2.7), although it seems that mCAF-3 cells are somewhat less sensitive to reovirus T3D^K^ compared to the other reoviruses (Figure 2). In the Western blot analyses and flow cytometry-based assay (Appendix A), less σ3 is detected in the T3D^K^-infected mCAF-3 cells, although p-MLKL levels are 2.7 times increased compared to untreated cells. This suggests that reovirus T3D^K^ is a relatively strong inducer of p-MLKL in murine cells.

The results in the murine cells demonstrate that not all reovirus strains induce cell death to the same extent in the various cell lines [26,29,30]. Studies performed with sialic acid binding reoviruses (T3SA+) versus non-sialic acid binding viruses (T3SA−) show that sialic acid binding induces a RIPK1-dependent and caspase-independent form of cell death (called necrosis in that study) not observed with the T3SA- reovirus [31]. While this could explain the induction of p-MLKL expression in jin-1 infected L929 cells, as this virus has enhanced binding to sialic acids [28], it cannot explain the difference in p-MLKL upregulation in T3D^K^-infected L929 cells compared to wt-R124 infected cells. These viruses have a single amino acid difference in their σ1 protein, at amino acid position 408 (Threonine in T3D^K^ and Alanine in both R124 and jin-1). 

In the study performed by Mohamed et al. [30], a mutation in S4 leading to amino acid change, W133R in σ3 of reovirus strain T3D^PL^ (a reovirus strain derived from the lab of Dr. Patrick Lee and currently in clinical trials, also known as Reolysin/Pelareorep [32]) is involved in the induction of RIG-I/IFN independent host genes and a faster replication phenotype. The three viruses used in the current study all possess W133 in σ3, suggesting that the observed differences in p-MLKL induction in mouse cells cannot be attributed to this amino acid. The difference can also not be explained by the other amino acid changes in σ3 present in jin-1 (see Appendix A), since these amino acid sequences are shared between T3D^K^ and wt-R124. 

Two mutations in M1 leading to amino acid changes in μ2 at positions 208 and 342 are thought to be involved in differences in plaque sizes. Reoviruses containing μ2P208 and μ2Q342 form larger plaques according to a different study by Mohamed et al. [33]. Our wt-R124 and jin-1 both contain the μ2P208 and μ2Q342 amino acids but strongly differ in plaque size in the HER911 cells; wt-R124 has larger plaques than jin-1 [28]. These two amino acids seem not to play a role in the induction of necroptosis in mouse cells in our hands since μ2 of reovirus T3D^K^ has both μ2S208 and μ2R342, and can induce p-MLKL in L929 cells, like jin-1, while wt-R124 is less efficient in p-MLKL upregulation. Recent studies by Boudreault et al. show that the interaction of μ2 with U5 small nuclear ribonucleoproteins (snRNPs) of the spliceosome is important for the regulation of necroptosis [34,35]. Knockdown of two proteins (EFTUD2 and SNRNP200) belonging to the U5 snRNP family resulted in a reduction of p-MLKL after infection of L929 cells with reovirus T3D^S^ (reovirus variant derived from the Lemay lab [36]). Although this interaction is of importance in some instances, in our hands it does not provide a full explanation for our findings that both jin-1 and T3D^K^ (µ2P208 and µ2S208 resp.) are stronger inducers of p-MLKL induction in L929 cells than the wt-R124 (µ2P208). This suggests that a combination of different factors is required. We did consider the other mutations in the different gene segments and introduced wt-R124 sequences into the T3D^K^ background in an attempt to change it to the less efficient necroptosis-inducing phenotype of wt-R124. We did not pursue this further when we realized that none of the reoviruses could induce necroptosis in the human cell lines. 

Recently, more knowledge has been gained about the human RIPK3:MLKL interactions and the differences between human and murine cells [37,38,39]. It is possible that in human cells, while the chemical cocktail TBZ can induce necroptosis, reoviruses can counteract the induction by interfering with the complex formation of RIPK3:MLKL. For instance, human cytomegalovirus (CMV) has been found to inhibit necroptosis in human cells [40]. In RIPK3-expressing human fibroblasts (HF cells), CMV infection induced p-MLKL, but CMV-infected HF cells were protected against cell death upon treatment with TBZ. These findings differ from our observations that reoviruses do not induce upregulation of p-MLKL in HT29 cells. This suggests that the potential repression of necroptosis in reovirus-infected human cells lies upstream of the phosphorylation of MLKL. The cell death as quantitated by the WST-1 assay (Figure 3C) shows that the cells are killed upon reovirus infection, with jin-1 most effectively inducing cell death in both the STA-et2.1 and the HT29 cell lines. These results suggest that in these human cell lines, other cell death pathways are initiated upon reovirus exposure. In HT29 cells the induction of apoptosis is probably an explanation for the cell death, as the induction of caspases 3/7 is increased in cells exposed to T3D^K^ and jin-1 even with a low MOI (Appendix A). 

Other viruses, like some poxviruses, express a kind of viral MLKL (vMLKL) protein, devoid of catalytic activity. This vMLKL binds to RIPK3 to prevent necroptosis in human U937 and HT29 cells [41]. The vMLKL proteins are encoded in the genomes of Cotia Poxvirus (COTV) and BeAn 58058 poxvirus (BAV) but not in some other poxviruses. Expression of vMLKL protein derived from BAV in HT29 cells resulted in the absence of p-MLKL due to the binding of vMLKL to RIPK3. 

The new insights into human versus murine MLKL:RIPK3 interactions show that it is difficult to predict how necroptosis is regulated in various species. The sequence homology of both RIPK3 and MLKL between humans and mice is about 60% [37] and there are differences in the interacting residues of mouse RIPK3:MLKL versus the human complex. In human cells, there is already an inactive open RIPK3:MLKL complex in the cytoplasm. Upon induction of necroptosis, this complex is activated and MLKL gets phosphorylated and assembles into oligomers that eventually trigger cell death. This does not explain, however, the phenomenon that we observe in the HER911 cells, where introducing exogenous RIPK3 already induces a high basal level of p-MLKL without causing cell death (Figure 4B). Even the addition of TBZ to these cells does not induce cell death (neither does it in the wt-HER911 cells). This is different from what was described for the HF cells, where reintroducing RIPK3 triggered cell death after exposure to TBZ [40]. Similarly, overexpression of RIPK3 in the immortalized human keratinocyte cell line HaCaT has been shown to induce p-MLKL and subsequent exposure to the chemical cocktail TSZ (containing SMAC-mimetic LCL-161 instead of BV6) still triggered cell death [42]. At this moment we cannot rule out that the transformed status of the HER911 cells by hAd5 contributes to the protection from cell death in the HER911hRIPK3 cells.

Taken together, our findings emphasize the complexities of translating results from mouse studies regarding ICD to human studies. This may limit the use of non-human models for necroptosis studies on oncolytic viruses especially if such studies have a translational intent.

## 4. Materials and Methods

### 4.1. Cell Lines and Viruses

Cell line HER911 was Ad5E1 transformed human embryonic retinoblasts, generated in our lab initially as helper cell line for replication-defective adenoviruses [43], VH10 cells (kindly provided by B. Klein) [44], STA-et2.1 (kindly provided by the LUMC Pediatrics department) [45], L929 (ATCC), HT29 (ATCC), U118 (ATCC), H1299 (ATCC) were cultured in high-glucose Dulbecco’s modified Eagle’s medium (DMEM; Gibco, Fisher Scientific, Landsmeer, The Netherlands), supplemented with 8% fetal bovine serum (FBS) (Biowest, VWR International, Amsterdam, The Netherlands) and antibiotics penicillin and streptomycin (pen/strep; Gibco, Fisher Scientific, Landsmeer, The Netherlands). STA-et2.1 cells are cultured on gelatin-coated dishes. Cell line BSRT7 expressing T7-RNA polymerase was provided by Professor K.K. Conzelmann (Ludwig-Maximilians-University Munich, München, Germany) and cultured in high-glucose DMEM, 8% FBS, pen/strep and 400 μg/mL G418 (Fisher Scientific, Landsmeer, The Netherlands) [46]. Mouse cancer-associated fibroblasts, mCAF-3, isolated from murine KPC3 tumor transplants, were obtained from the Department of Gastroenterology and Hepatology at the LUMC and cultured in DMEM/F12 medium (Gibco), supplemented with 8% FBS. All cells were cultured in an atmosphere of 5% CO_2_ at 37 °C and confirmed negative for the presence of mycoplasma by regular testing (MycoAlert Mycoplasma Detection Kit, Lonza Benelux, Breda, The Netherlands).

The wt-T3D virus strain R124 [28] was isolated in our lab from reovirus T3D stock VR-824 (ATCC) by two rounds of plaque purification in HER911 cells and further propagated on HER911 cells. In the text, R124 is referred to as wt-R124. The jin-1 mutant reovirus was obtained by bioselection of reovirus T3D stock VR-824 on U118MG cells as previously described [28] and further propagated on HER911 cells.

Plasmid-derived reovirus T3D^K^ [47] was generated as previously described [48]. In short, the reovirus-genome-containing plasmids (obtained from Addgene) were introduced with TransIT-LT1 transfection reagent (Mirus, Sopachem BV, Ochten, The Netherlands) in BSRT7 cells (in 6-well plates). The BSRT7 cells were lysed by 3 cycles of freeze-thawing 72 h post transfection. After removal of the cell debris by centrifugation, half of the lysate was added to HER911 cells (in 6-well plates) in the presence of fresh DMEM-2%FBS. At the first signs of CPE, viruses were harvested from the cells and further propagated on HER911 cells. At passage 3, reovirus T3D^K^ was grown in five T75 flasks HER911 cells for cesium chloride (CsCl) purification. 

The purification procedure was adapted from Berard et al. [49] with some modifications. In short, reovirus-infected cells were harvested when cells are ±40% in cytopathogenic effect (CPE) and cells were separated from the medium by centrifugation (10 min, 805× *g*), resuspended in 2 mL HO buffer (10 mM Tris.Cl pH = 7.8, 250 mM NaCl and 1 mM MgCl_2_) and transferred to −80 °C for one freeze-thaw cycle. Triton X100 (0.1%) and 50 U benzonase (Merck Life Sciences, Amsterdam, The Netherlands) were added to the thawed cell suspension and incubated for 15 min at 4 °C, followed by 15 min 37 °C and 10 min at 4 °C. Pre-purification was performed by two sequential Vertrel XF (Halotec CL10, FenS, Goes, The Netherlands) extractions before loading on a 1.20–1.44 g/mL CsCl block gradient for final purification. Centrifugation was performed in a Beckman Coulter Optima XE-90 ultracentrifuge using an SW41-Ti swing-out rotor at 69,000× *g* for 14 h at 10 °C. CsCl was replaced with reovirus storage buffer (RSB; 10 mM Tris.Cl pH7.5, 150 mM NaCl and 10 mM MgCl_2_) by Amicon filtration (Amicon Ultra 100K, Merck-Millipore, Amsterdam, The Netherlands). Purified reoviruses were stored at 4 °C until further use.

The multiplicity of infection (MOI) mentioned in this manuscript is based on the reovirus titers, determined by plaque assays on HER911 cells.

### 4.2. Generation HER911 Cells Expressing Human RIPK3

To clone *human RIPK3* in lentiviral vector pLV.CMV.IRES.Puro [50], oligo (dT) synthesized cDNA of THP-1 cells (ATCC) was used as input for PCR with primers hRIPK3_For 5′-ccaccatgtcgtgcgtcaagttatggc and hRIPK3_Rev 5′-ttatttcccgctatgattataccaaccct. The human RIPK3 PCR product was inserted by traditional ligation, in the Eco32I site of pLV.CMV.IRES.Puro to create pLVhRIPK3. The resulting plasmid was sequenced to confirm the presence of the correct *human RIPK3* sequence. Lentivirus LV-hRIPK3 was generated in 293T cells as described previously [51]. HER911 cells were transduced with LV-hRIPK3 and three days post infection, cells were transferred to puromycin (0.6 µg/mL, InvivoGen, Bio-Connect, Huissen, The Netherlands) containing DMEM-8% FBS. Cells were pooled to generate a polyclonal cell line, HER911hRIPK3 PC100. Expression of human RIPK3 was confirmed by RT-qPCR and Western blot.

### 4.3. Western Analysis

For detection of proteins by western analysis, infected cells (1*10^5^ per well) in 24-well plates were lysed at indicated time points with RiPA buffer (Pierce, Fisher Scientific, Landsmeer, The Netherlands), supplemented with protease inhibitors (Complete mini tablets, Roche Diagnostics, Almere, The Netherlands) and 50 U/mL Benzonase (Merck Life Sciences). Before use, 10 mM NaF (Merck Life Sciences, Amsterdam, The Netherlands) was added to the RiPA buffer. Samples were kept on ice to prevent the degradation of proteins. Immediately after lysis of the cells, 4× sample buffer was added (final concentration 2% SDS, 0.1 M DTT, 0.1% Bromophenol blue and 10% glycerol) and aliquots (25 µL) were stored at −80 °C until used for loading on a 10% polyacrylamide-SDS gel (20 µL per sample). Proteins were transferred using the trans-blot turbo system from the gel to a nitrocellulose membrane (Bio-Rad Laboratories, Veenendaal, The Netherlands). The marker used for molecular weight marker was All Blue Precision plus (Bio-Rad Laboratories, Veenendaal, The Netherlands). For phosphorylated proteins, the blot was blocked with 5% BSA in TBS (10 mM Tris.Cl, 150 mM NaCl) without Tween-20. Wash steps were performed with TBS supplemented with 0.1% Tween-20. To separately detect p-MLKL and total MLKL, both antibodies were diluted in Immuno Booster Solution 1 (Takara Bio Europe SAS, Saint-Germain-en-Laye, France). A mixture of the corresponding secondary antibodies was diluted in Immuno Booster Solution 2 (Takara Bio Europe SAS, Saint-Germain-en-Laye, France). Primary antibodies used: Mouse anti-MLKL, phospho S345 (ab196436, 1:1000); human anti-MLKL, phospho S358 (ab187091, 1:1000; Abcam, ITK Diagnostics, Uithoorn, The Netherlands); Rat anti-MLKL, clone 3H1 (MABC604, 1:2000; Millipore, Merck-Millipore, Amsterdam, The Netherlands); reovirus anti-σ3, 4F2 (1:3000; Developmental Studies Hybridoma Bank, University of Iowa, Iowa City, USA); rabbit anti-human RIPK3 (#10188, 1:1000; Cell Signaling Technology, Leiden, The Netherlands); mouse anti-Vinculin (Clone hVIN-1, V9131, 1:10,000; Merck Life Sciences, Amsterdam, The Netherlands). Secondary antibodies: IRDye 800CW Donkey anti-R IgG; IRDye 680RD Donkey anti-M IgG; IRDye 680RD Goat anti-Rat IgG (all 1:5000; LiCor, Westburg, Leusden, The Netherlands). Signals were detected with the Odyssey Clx imaging system. To visualize staining and calculate ratios p-MLKL/MLKLtot, Image Studio Lite software was used (LiCor, Westburg, Leusden, The Netherlands). 

As a control for the induction of p-MLKL, the TBZ cocktail was used, which consists of TNFα (50 ng/mL; InvivoGen Europe, Toulouse, France), SMAC mimetic BV6 (0.1 µM; Abmole, S.A. ForLab, Brussel, Belgium) and Z-VAD-FMK (20 µM; Bachem AG, Budendorf, Switzerland) [52].

### 4.4. Cell Viability Detection

WST-1 reagent (Roche diagnostics, Almere, The Netherlands) was used to determine the viability of cells after reovirus infections. Cells were mock-infected or infected with wt-R124, T3D^K^ and jin-1 with an MOI 10 (three independent infections). Three days post-infection, WST-1 reagent was added according to the manufacturer’s instructions. To determine the level of cleaved product, the optical density was measured at 450 nm after incubation at 37 °C. Viability calculations were normalized to mock-infected cells (set at 100%).

### 4.5. Human RIPK3 RT-qPCR

RNA of cell lines was used to generate cDNA, using an oligo(dT)20 primer (Enzo Life sciences, Raamsdonksveer, The Netherlands) and Superscript II reverse transcriptase (Fisher Scientific, Landsmeer, The Netherlands). Expression of human RIPK3 was measured by real-time quantitative PCR (qPCR), using iQ SYBR Green Supermix (Bio-Rad Laboratories, Veenendaal, The Netherlands) on a CFx connect thermocycler (Bio-Rad Laboratories, Veenendaal, The Netherlands). Primers (all from Merck Life Sciences, Amsterdam, The Netherlands) for hRIPK3 qPCR: Forward 5′-gctacgatgtggcggtcaagat and Reverse 5′-ttggtcccagttcaccttctcg. Primers for human GAPDH amplification used for normalization: Forward 5′-catcatccctgcctctactg and Reverse 5′-ttggcaggtttttctagacg. Thermocycler program: initial denaturation 3 min at 95 °C, followed by 40 cycles of 10 s at 95 °C and 30 s at 60 °C, followed by melt curve detection. Relative hRIPK3 expression was calculated using the 2^(−ΔΔCt)^ method, using GAPDH as an internal reference [53]. Background expression was calculated by assuming a human RIPK3 Ct threshold of 40 in cell lines that showed no Ct value (40 cycles are used for the PCR) and with the average GAPDH Ct values of the corresponding cell lines.

### 4.6. Cell Death Detection with Propidium Iodide

Cells were treated or not with TBZ as stated before in 24-well format in 400 µL DMEM 8% FBS. After 24 h, 100 µL of PI-Hoechst solution (in medium) was added to achieve a final concentration of 8 µM Hoechst 33342 (Fisher Scientific, Landsmeer, The Netherlands) and 0.8 µM propidium iodide (PI) (Fisher Scientific, Landsmeer, The Netherlands) in a total volume of 500 µL. The medium was removed after 20 min of incubation at room temperature (dark) and replaced with 1% PFA in PBS for a quick fixation step (3 min). Immediately after removing the 1% PFA solution, pictures were taken with a Leica DMi8 microscope connected to an EL6000 external light source. All pictures were taken with a magnification of 10×. PI- and Hoechst-positive cells, from the same area, were counted with Fiji using macros as described in Appendix A. The fraction of PI-positive cells was calculated by dividing the number of PI-positive cells by the number of Hoechst-positive cells in the selected areas.

## Figures and Tables

**Figure 1 ijms-24-02320-f001:**
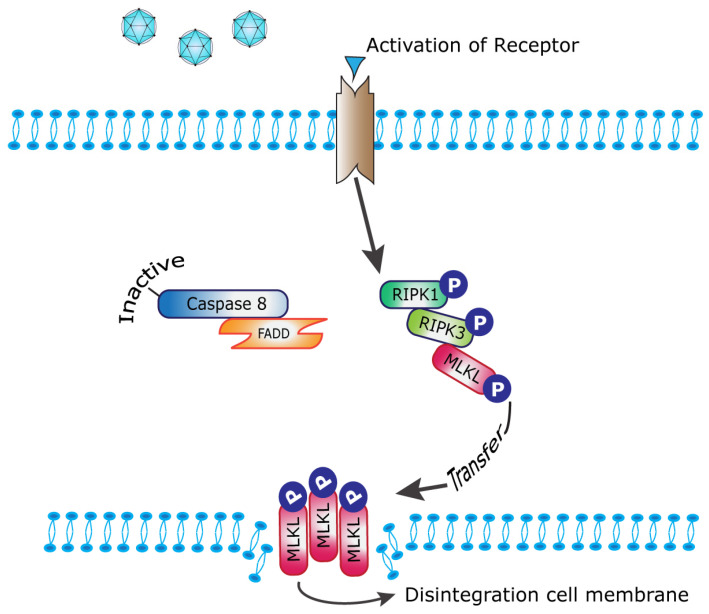
Simplified scheme of necroptosis pathway. Upon activation of the membrane receptor (often by TNF-family members), members of the receptor-interacting protein kinases (RIPK1 and RIPK3) are phosphorylated. When caspase 8 is in an inactive state, p-RIPK3 recruits and phosphorylates mixed-lineage kinase domain-like protein (MLKL). After the oligomerization of p-MLKL, the complex ends up in the cell membrane and causes its disintegration. This results in the release of cellular contents into the environment. In cell culture, necroptosis can be induced by the treatment of cells with a cocktail of TNFα, SMAC mimetics and Z-VAD. This cocktail activates the receptor, inhibits caspase activation to prevent apoptosis and blocks ubiquitylation of RIPK1 which pushes the cells towards necroptosis.

**Figure 2 ijms-24-02320-f002:**
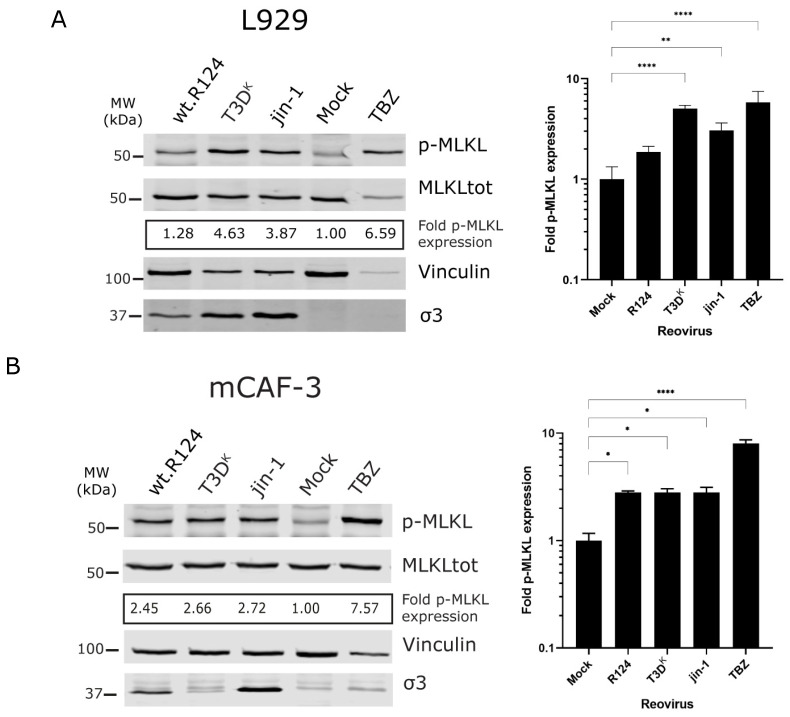
Expression of p-MLKL in reovirus-infected mouse cells. (**A**) L929 cells or (**B**) mouse cancer-associated fibroblasts (mCAF-3) were exposed to wt-R124, T3D^K^ and jin-1 (MOI = 10) or mock-treated. As a positive control, cells were treated with TBZ. Lysates were made 30 h post treatment. The fold induction of p-MLKL expression was calculated by quantifying the p-MLKL/MLKLtot ratio, normalized to the mock-treated cells (corresponding p-MLKL and MLKLtot color panels, see Appendix A). Vinculin was used as a loading control and σ3 as a control for the presence of reovirus in cells. (**A**) One representative Western blot is shown out of 2 independent experiments. The graph represents the fold induction of p-MLKL expression from two separate experiments and 5 independent wells for reovirus infections and 4 independent wells for both mock- and TBZ-treated cells. Statistical significance was determined by one-way ANOVA with Bonferroni’s multiple comparisons: ** *p* = 0.0033, **** *p* < 0.0001. (**B**) One representative Western blot is shown out of 2 independent experiments. The graph represents the fold p-MLKL induction from two independent experiments. Statistical significance was determined by one-way ANOVA with Bonferroni’s multiple comparisons: * *p* = 0.0136, **** *p* < 0.0001. (**C**) L929 cells and mCAF-3 cells were mock-treated or exposed to wt-R124, T3D^K^ and jin-1 (MOI = 10). Viability was assessed by WST-1 assay, 3 days post-infection, from 3 independent infected wells. Error bars represent SD from the triplicates. Statistical significance was determined by two-way ANOVA with Bonferroni’s multiple comparisons: **** *p* < 0.0001.

**Figure 3 ijms-24-02320-f003:**
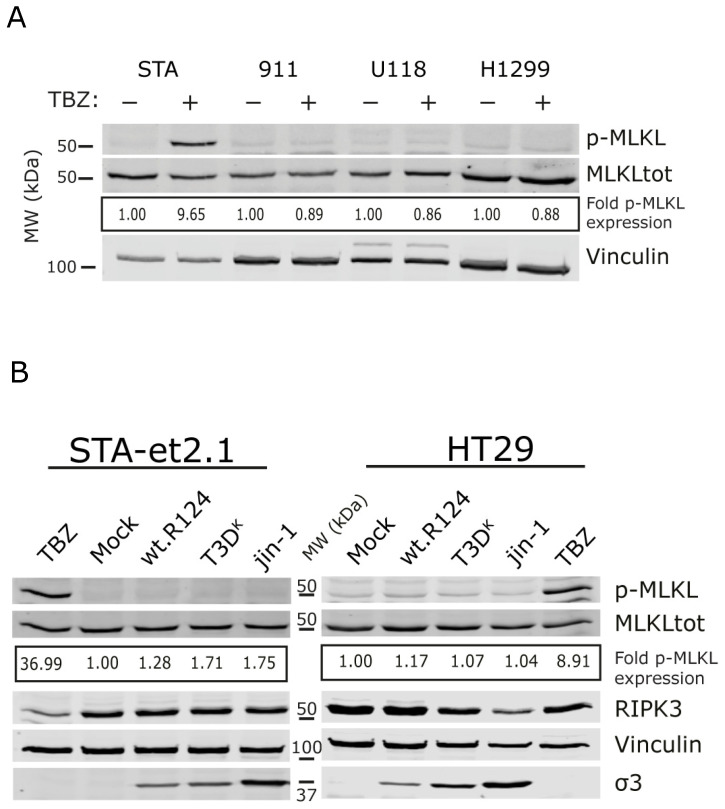
Expression of p-MLKL in reovirus-infected human STA-et2.1 and HT29 cells. (**A**) Expression of p-MLKL expression after induction by the TBZ cocktail in different human cell lines. Cells were treated (+) or not (−) with TBZ. Lysates were made 24 h post treatment. The fold induction of p-MLKL expression was calculated by quantifying the p-MLKL/MLKLtot ratio, normalized to the mock-treated cells (corresponding p-MLKL and MLKLtot color panels, see Appendix A). Vinculin was used as a loading control. STA stands for STA-et2.1 cells, 911 for HER911 cells. (**B**) STA-et2.1 and HT29 cells were exposed to wt-R124, T3D^K^ and jin-1 (MOI = 50) or mock-treated. TBZ-treated cells were included as a positive control. Lysates were made 30 h post treatment. The fold induction of p-MLKL expression was calculated by quantifying the p-MLKL/MLKLtot ratio, normalized to the mock-treated cells (corresponding p-MLKL and MLKLtot color panels, see Appendix A). Vinculin was used as a loading control and σ3 as a control for the presence of reovirus in cells. RIPK3 is included to show that cell lines express RIPK3. One representative Western blot is shown out of three independent experiments. (**C**) HT29 and STA-et2.1 cells were mock-treated or exposed to wt-R124, T3D^K^ and jin-1 (MOI = 10). Viability was assessed by WST-1 assay, 3 days post infection, from 3 independent infected wells. Error bars represent SD from the triplicates. Statistical significance was determined by two-way ANOVA with Bonferroni’s multiple comparisons: ** *p* = 0.0019, **** *p* < 0.0001.

**Figure 4 ijms-24-02320-f004:**
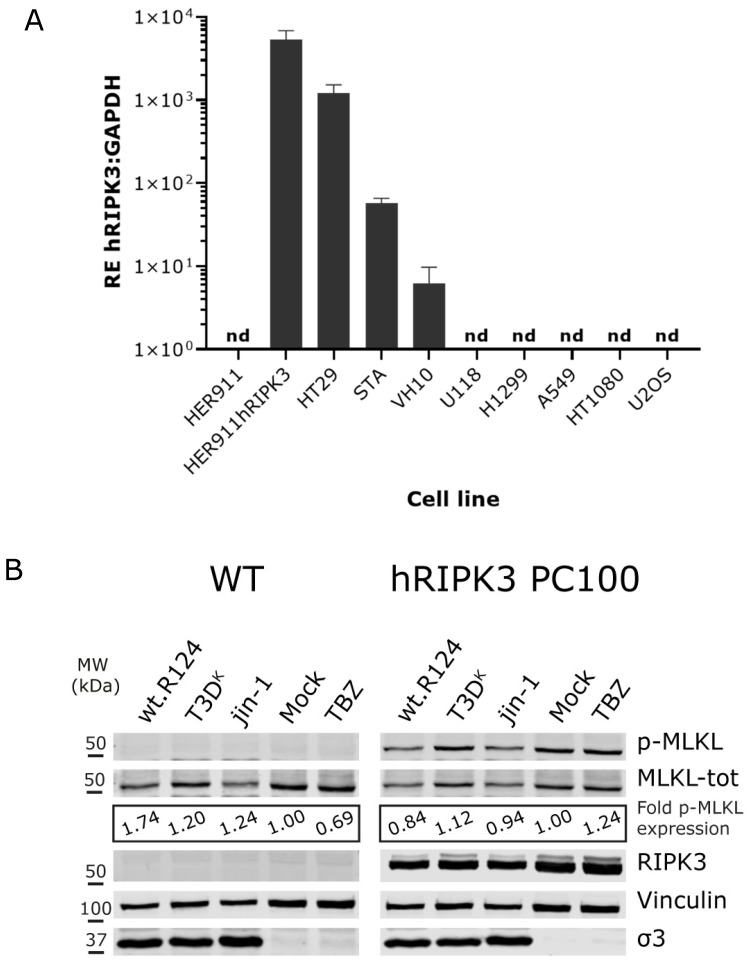
Forced expression of RIPK3 cDNA in HER911 cells leads to constitutive p-MLKL expression, but not to cell membrane disruption. (**A**) Normalized RIPK3 mRNA expression per cell line. RNA from indicated cell lines was used as input for Reverse Transcriptase qPCR with human RIPK3 primers and normalized to GAPDH. HER911hRIPK3 cells are human RIPK3 transduced HER911 cells (polyclonal). “nd”: no Ct values detected. Error bars represent SD from technical triplicates. (**B**) HER911 (WT) and HER911hRIPK3 PC100 (hRIPK3 PC100) transduced cells were exposed to wt-R124, T3D^K^ and jin-1 (MOI = 10) or mock-treated. TBZ-treated cells were included to check if the addition of human RIPK3 in 911 cells sensitizes these cells to the induction of p-MLKL. Lysates were made 30 h post treatment. The fold p-MLKL expression was calculated by quantifying the p-MLKL/MLKLtot ratio, normalized to the mock-treated cells (corresponding p-MLKL and MLKLtot color panels, see Appendix A). RIPK3 antibody is used to show the expression in the HER911hRIPK3 PC100 cells. Vinculin was used as a loading control and σ3 as a control for the presence of reovirus in cells. (**C**) The fraction of PI-positive cells after treatment with TBZ. Cells were treated or not with TBZ for 24 h before staining with propidium iodide (PI) and Hoechst 33342. Immediately after staining, pictures of PI and Hoechst signal were taken (for an example of the staining see Appendix A), and positive cells were counted with Fiji from three different areas in two duplicate wells. The fraction of PI-positive cells was calculated by dividing the number of PI-positive cells by the number of Hoechst-positive cells. Error bars represent SD from the three different areas. Statistical significance was determined by 2-way ANOVA with Bonferroni’s multiple comparisons: ** *p* = 0.0032, *** *p* = 0.0008, **** *p* < 0.0001.

## Data Availability

The raw data of the current study are available from the corresponding author upon request.

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
