# Peer review of "Reovirus Type 3 Dearing Variants Do Not Induce Necroptosis in RIPK3-Expressing Human Tumor Cell Lines"

_ijms, 2023, doi:10.3390/ijms24032320_

Round 1
Reviewer 1 Report
In this manuscript, the authors looked at necroptosis in the context of reovirus infection. They essentially measured phosphorylation of MLKL, which is fine to start with. This is also interesting since there are relatively few manuscripts on reovirus-inducted necroptosis and much remains to be done. Essentially, their objective was to examine this aspect in human cancer cell lines, considering the oncolytic activity of reovirus and the importance of necroptosis in immune aspect of the antitumoral response. Altogether, this makes sense and is of importance, although I found that the actual objectives of the work were somewhat unclear at the beginning and a little lost in the manuscript. They also introduce the question of differences in necroptotic potential of three reovirus variants but, at the end, the impact of this last aspect is not very well established.
Although the goal of the manuscript, and its putative impact, appears of importance to me, my enthusiasm was dampened by the data presented that do not add much to our understanding and do not appear that convincing as they are presented. I found it quite striking that the discussion section is almost twice as long as the text of the results section; it does appear as highly speculative with few supporting data at this point; I guess the authors could modify the discussion to go to the point: what do the results show that convincingly represent a useful addition to the scientific literature?
You will find below more specific points that I believed should at least be addressed.
•The work is mostly based on immunoblotting (Western) analysis using LiCor which is quite a good way for quantitative analysis. However, it is unclear how many times the experiments were repeated if they actually were? Since relatively small differences are presented and discussed in some experiments, this is of importance. Ideally, mean values (or mean differences) should be presented with error bars (or something similar, such as a detailed description in the text of values obtained). Also, there is apparently a high background with numerous bands on these western blots without much explanation, or I may be missing something? How do the authors established which band is actually p-MLKL/MLKL (they comigrate if I understand it well), this should be at least mentioned.
•Paragraph starting at line 92: It is a good idea to compare viral variants but the justification presented is quite limited. One important aspect is the interferon response that is known to be linked to necroptosis. The authors should indicate if they looked (or know) if their strains do exhibit differences in interferon induction. Since there is some literature on the viral genes and polymorphisms involved, predictions could at least be made.
•Line 104: It seems surprising that overexpression of p-MLKL in HER911 cells does not cause cell death. Is it possible that a subpopulation of resistant cells was selected in the process? Please discuss.
•Figure 1: As I understand it, the point here is to show that there are differences in necroptosis efficiency among the three viruses and the two cell lines. With just one data, and such small differences, this is far from convincing. Also, the p-MLKL/MLKL ratio is used, which is good, but how about the amount of viral protein? If you take into account the various amounts of sigma3, it here such a difference between the viruses and the cell lines? Is it statistically significant, or at least is it reproducible upon at least three separate infection? The number of positive cells is not that useful, considering that it obviously does not correlate with efficiency of infection. Also, is there a difference in the kinetics if one look at different time points?
•Figure 4: It will be interesting to have L929 run in parallel. The pMLKL level in mock cells appear much lower than it was for L929, but it is not the same antibody. It is still quite surprising that induction of 1.75 is seen at least for one virus, higher than in L929 cells, this is overall a little confusing, although I believe that the conclusion is right at the end it should be better explained. Also, the order of the virus is different on the two gels, that adds to the confusion.
•Figure 5: Is there a reason to use RT-qPCR for this single experiment rather than western blotting as the rest of the manuscript? Was the GAPDH values similar in all samples? Otherwise, does that possibly affect the comparison? I am not sure that I understand the description of Materials and Methods: “Background expression was calculated by assuming a hRIPK3 Ct threshold of 40, and with the average GAPDH Ct values of all samples.” Do you mean in “each” sample? Overall, why not a western as the rest of the work? Please justify this choice of technique at this point in the paper.
•Figure 6: Overexpression of RIP3 was nicely achieved in HER911 cells but the end result is a constitutively high level of p-MLKL with no other apparent impact. I fail to really see the interest in this result; the authors have to convince me. Is there any way to take advantage of this observation whatsoever? Or am I missing something?
Reviewer 2 Report
The authors investigate the induction of necroptosis by variants of the reovirus prototype strain Type 3 Dearing (T3D). They confirm previous work demonstrating phosphorylation of MLKL (indicative of necroptosis) in murine cell lines following reovirus infection. In contrast, human cell lines STA-et2.1 and HT29 did not exhibit p-MLKL following infection, despite being competent to induce p-MLKL following treatment with the positive control TBZ cocktail, and despite evidence of reovirus infection (assessed via expression of the reovirus structural protein s3). Forced expression of the MLKL-phosphorylating kinase RIPK3 in a human cell line that does not normally express this protein, HER911, led to constitutive phosphorylation of MLKL, but no increase in phosphorylation following infection. Interestingly, this constitutive expression of RIPK3 did not lead to an increased loss of cell viability following TBZ treatment.
Overall, the data is relatively clear and generally well-controlled, and the manuscript is well-written. The main conclusions are a good reminder to the field that not all cell types and species of origin function similarly, and has implications for the use of reovirus as an oncolytic virotherapy. However, there are some components missing that would allow for a deeper interpretation of the data. At the core, the authors draw conclusions about differences between reovirus variants that are not warranted by the data that is presented. No replicates are shown for many of the experiments, and while relative levels of pMLKL are provided, there is no indication of the variability of these results, or the statistical significance of the differences between variants. The differences observed may very well be within the natural experimental variability; thus, none of the comparative statements made (or commented on in the discussion) are justified by the data. A second major concern is that relative levels of pMLKL may not necessarily correlate with differences in cell viability – outside of figure 6 (which is irrelevant for reovirus-induced cell death), no measurements were made of cell viability, so any claims that there are differences in “necroptosis induction” are not warranted. The data would be much better supported with measurements of cell viability (via propidium iodide staining or otherwise) following infection. This would also help inform experiments where reovirus infection does not induce pMLKL – are these cells dying via other mechanisms, or is cell death inhibited in these cells? Ideally, caspase activity would also be measured, as it may be that in these cell types, reovirus may induce more of an apoptotic form of cell death. Given the role of caspase-8 in inhibiting necroptosis, an additional, potentially informative experiment would be to treat these cells with a caspase-8 inhibitor during infection, to determine whether this would restore the capacity of reovirus to induce p-MLKL and necroptosis.
Minor concerns:
1) In Figure 2C, what do the error bars represent? Is this the variability of the two separate infections? Or of separate samples of one infection (i.e. triplicate wells of cells)?
2) In Figure 4A, it would be helpful to maintain the same order of samples for the immunoblots.
3) In Figure 5, mRNA levels are measured, but these do not always fully correlate with protein levels; an immunoblot comparing RIPK3 protein levels amongst these cell lines would be informative.
Reviewer 3 Report
In this manuscript, Van Den Wollenberg et al investigated the impact of mammalian orthoreovirus (herein, reovirus) on necroptosis of human cancer cell lines. The authors conclude that reovirus does not induce necroptosis (based mainly on the phosphorylation of MLKL) in these cells in comparison to mouse cells (L929, mCAF-3), and that care must be taken when using murine models of necroptosis to study oncolytic viruses. The research question is very interesting and this reviewer particularly like the angle (comparison between mouse and human cells) and the comparison of multiple reovirus strains. However, some drawbacks and questions arose during the reading of the manuscript that the authors should address.
General comments
The manuscript requires more work (rewording, clearer ideas and hypothesis, more replicates, etc). Generally speaking, the conclusions and the link between the different section are not always clear. For example, results sections 2.2 and 2.3 do not even finish with a concluding statement or a clear take-home message. Some cells lines are not adequately introduced, and the way some things are called is confusing. Figure legends contains an anormal amounts of details (antibody number, type, and usage of confusing nomenclature such as IRDye DoαR 800CW, for example). Please simplify the figure legend and make sure it does not include things that belongs to the material and methods section, such as antibody type, number, etc.). I would suggest the authors rework the manuscript to make it clearer.
Major critiques
The authors based their finding that reovirus induce necroptosis in murine cells on fibroblasts (L929, and cancer-associated fibroblasts, mCAF-3). However, in human cell lines, they directly tested tumor cells lines (STA-et2.1, HT29, H1299, U118, HER911). This hampers to correctly conclude that human cells are resistant to reovirus-mediated necroptosis, as these are cancer cells, in comparison to fibroblasts for murine cells. Thus, the difference could only reside in the nature of cells tested (cancer vs fibroblast). The inclusion of mouse cancer cell lines (CT26, CT36, 4T1, etc), or human fibroblasts, is required to clearly confirm the hypothesis of the authors that murine cells are susceptible to reovirus-mediated necroptosis and human cells are not, and that it is not that cancer cells are resistant, and fibroblasts are susceptible, no matter the specie.
Figure 2A. Please explain why your mock infected L929 cells present such a high level of pMLKL. This is in complete opposition with previously published papers showing very low to no levels of pMLKL at basal level in L929 (Boudreault et al, Viruses, 2022; Berger et al, Journal of Virology, 2017; Koehler et al, PNAS, 2017; Koehler et al, Cell Host and Microbe, 2021) and cast doubt about the results presented here. Same things with Figure 2B, where mock mCAF-3 cells also shown a significant band for pMLKL. Also, in Figure 2A, there are drastic differences in total MLKL levels (TBZ condition being very low) and vinculin loading controls (TBZ being very low, and WT.R124 and mock being higher). Please run this blot again and ensure a more constant loading of the different protein samples on the gel.
Figure 2C-4C. Only one experiment is presented with two different infections; it is thus a technical replicate. This is not sufficient to draw a solid conclusion; please run the experiment in biological triplicate (three times) and add statistics to the analysis.
Figure 4A. Why are the WB loaded in a completely different way than Figure 2A? This is confusing for the reader. Please rerun the blots and make sure that the samples a loaded in a coherent fashion for the reader to understand the figure. The three viruses (WT.R124, T3D.Kob, and Jin), Mock, and the TBZ positive control is a logical way to present your result and should be preserved throughout the manuscript.
Figure 5. Only one experiment is presented with technical triplicate; this is not sufficient to draw a conclusion. Include biological triplicates in this experiment and run statistical analyses on the data. Moreover, the authors monitored RIPK3 expression through its mRNA levels; this is a simplistic way to look at RIPK3 activity, as the mRNA levels is not correlated with the protein levels (even more so in cancer cells). A WB should be included to monitor the protein level of RIPK3, which is what the authors are trying to monitor indirectly through mRNA levels.
It is stated throughout the manuscript that lysates are made 30 h (or sometimes 24 h) after treatment, even with the positive control TBZ. It is well known that, at least for L929 cells, cells die by necroptosis in approx. 6h-8h using only zVAD-fmk 100 uM/TNFα 25 ng/mL (Boudreault et al, Viruses, 2022; Koehler et al, PNAS, 2017; Koehler et al, Cell Host and Microbe, 2021). Why use such a long incubation time? Are the cells not already dead by the time you harvest them?
The authors do not describe how they measure the viral titers of their different viruses before infection. It is very important, as they compare different viruses and different cell lines altogether. Virus titers measured in one cell line might not be the same in another cell line, and this might hamper adequate comparison done using the three different viruses.
Minor critiques
Please revise the manuscript for words separated by a hyphen. I detect numerous instances of them (list not exhaustive):
Onco-lytic (line 44); path-way (line 69); ex-pressed (line 84); intro-duction (line 103); in-duction (line 124); per-centage (line 127); Anti-body (line 114); ex-pression (line 216); over-all (line 224); up-regulate (line 236); in-duction (line 237); up-regulation (line 254); vi-ruses (line 259); com-plex (line 280); de-void (line 288); ex-pression (line 290); ex-plain (line 301); de-scribed (line 335); af-ter (line 408).
The usage of T3D.Kob is specific to this manuscript, please use either the nomenclature T3DK (Lanoie et al, Virus Research, 2018; Després et al, Viruses, 2022; Boudreault et al, Nucleic Acids Research, 2022) or T3DTD (Mohamed et al, Journal of Virology, 2019; Mohamed et al, Plos Pathogen, 2020; Mohamed et al, Journal of Virology, 2020) which are already widely used in the field.
Please add a reference to line 53.
Line 74: remove the , between in both and apoptosis.
In the introduction, the authors do not discuss other very-well characterized sensors of necroptosis (ZBP1, TRIM). Notably, it was recently shown that inhibiting RIPK1 is not sufficient to completely abrogate reovirus-mediated necroptosis in L929 cells (Boudreault et al, Viruses, 2022). These elements should be added to the introduction (or the discussion). Figure 1 should be corrected to reflect that there are three different sensors for necroptosis and not only RIPK1.
Line 123: please define TBZ in the result section when it is first mentioned.
I don’t understand why figure 3 and 4 are not grouped together; they are discussed in the same section of the result section and figure 4 stems from figure 3. I think they should be grouped together. The same is also true for figure 5 and 6.
At the beginning of section 2.3, the authors focussed their attention of HER911 because they do not express any RIPK3 protein. What is the basis of that statement? And what are HER911 cells? They are not adequately introduced. Also, please keep your nomenclature constant throughout the manuscript, and replace the 911 by HER911, as it confusing to have both.
Please avoid using hRIPK3 to describe RIPK3 in human cells. It is confusing to read statements such as "Since the HER911 cells do not express detectable hRIPK3 expression"; these are human cells and as such, they can only express the human form of RIPK3. Refer to human RIPK3 of murine RIPK3 and avoid wordings where it is suggested that cells could either express human or murine RIPK3, as it is impossible. Moreover, please use the correct nomenclature (RIPK3) throughout the manuscript (and not RIP3).
Figure 6B. Why are the results presented in a logarithmic scale? A percentage of dead cells should be presented in a linear scale.
Line 256-257. Please write the complete amino acid names and not just the three-letter abbreviation.
Line 257-258. "In the study done by Mohamed et. al. [27] a mutation in S4 leading to amino acid change, W133R in T3DPLσ3 is responsible for the induction of RIG-I/IFN independent host genes and a faster replication". Please explain T3DPL, use the correct nomenclature (T3DPL), and review this sentence as it problematic.
Line 265-272: The authors discuss about μ2 and the amino acid at position 208. Its was previously shown that this amino acid is responsible for the different impact on cellular AS (Boudreault et al, NAR, 2022), and that μ2 notably interacts with EFTUD2, PRPF8, and SNRNP200 from the U5 snRNP of the spliceome. A recent follow-up of that study (Boudreault et al, Viruses, 2022) showed that EFTUD2 and SNRNP200 are required to induce necroptosis during reovirus infection. This should be discussed and included in the discussion, as this suggest μ2 might have an important role in necroptosis regulation through its interaction with EFTUD2 and SNRNP200.
Line 292-293, the authors state: "So far, it is not known whether reoviruses encode protein sequences in one of their segments that mimic part of the human MLKL protein". This is easily verifiable as amino acids sequences for human MLKL and reovirus are known; the authors should either verify this hypothesis, or not mention this possibility.
The authors fail do discuss the importance of necroptosis during the oncolytic activity of reovirus; what is know about the induction of necroptosis in animal models of cancer? 2D cell culture models of cancer are very simplistic, and it should be discussed whether these findings will really translate to animal models and models where the immune system is active.
Line 372: Please correct 1*105 per well to 1*105 per well.
Please explain why you fixed cells in the cell death detection with propidium iodide section of Materials and Methods. As PI is a dye that required live cell to be correctly effluxed outside the cell when cells are alive, wouldn’t it be better to directly monitor live cells?
Round 2
Reviewer 1 Report
Overall, the authors have made important changes to their manuscript in order to answer my previous comments. The major point is the addition of bar graphs to show quantitation on more than one experiment for immunoblots, although the number or biological replicates are limited the results are quite clear. That was missing from the previous version and was the most important drawback, from my point of view.
Although, there are still aspects that could have been better assessed, such as comparing parental and cancer cells from the same species, and confirming that viral replication is similar in the human cells compared to L929 by kinetic analysis of viral titers, I believe that the current version is still adequate for publication and will be a useful addition to the scientific literature.
It is difficult to read from the annotated pdf file, but it will be important for the authors to verity before final publication that all figure numbers and corresponding legends are correct. For example, there seems to be two figures labeled as figure 4.
Author Response
Overall, the authors have made important changes to their manuscript in order to answer my previous comments. The major point is the addition of bar graphs to show quantitation on more than one experiment for immunoblots, although the number or biological replicates are limited the results are quite clear. That was missing from the previous version and was the most important drawback, from my point of view.
Although, there are still aspects that could have been better assessed, such as comparing parental and cancer cells from the same species, and confirming that viral replication is similar in the human cells compared to L929 by kinetic analysis of viral titers, I believe that the current version is still adequate for publication and will be a useful addition to the scientific literature.
It is difficult to read from the annotated pdf file, but it will be important for the authors to verity before final publication that all figure numbers and corresponding legends are correct. For example, there seems to be two figures labeled as figure 4.
Response: We thank the reviewer for the time and constructive suggestions, which improved our manuscript. As was suggested, we corrected the figure labeling below the figures and in the text (in the manuscript as well as in the supplementary figures). We detected some inconsistencies in nomenclature and corrected these as well.
Reviewer 2 Report
The manuscript is improved with the inclusion of more information regarding replicates and statistical analysis. One minor point is that the results section describing overexpression of RIPK3 should now refer to figure 4, not figure 3. There are still some concerns about the relative lack of mechanistic insight into the cell death pathways elicited by reovirus in human cell lines, in the absence of a strong necroptotic signal, which diminishes the significance of (and therefore the enthusiasm for) the work.
Author Response
The manuscript is improved with the inclusion of more information regarding replicates and statistical analysis. One minor point is that the results section describing overexpression of RIPK3 should now refer to figure 4, not figure 3. There are still some concerns about the relative lack of mechanistic insight into the cell death pathways elicited by reovirus in human cell lines, in the absence of a strong necroptotic signal, which diminishes the significance of (and therefore the enthusiasm for) the work.
Response: We thank the reviewer for the time and constructive suggestions, which improved our manuscript. As was suggested, we corrected the figure labeling below the figure and in the text (in the manuscript as well as in the supplementary figures). We detected some inconsistencies in nomenclature and corrected these as well.
We hope that our future experiments will yield more knowledge on the mechanisms involved in reovirus oncolysis and their induction of ICD.